# Genomic Adaptations of *Saccharomyces* Genus to Wine Niche

**DOI:** 10.3390/microorganisms10091811

**Published:** 2022-09-09

**Authors:** Estéfani García-Ríos, José Manuel Guillamón

**Affiliations:** 1Department of Food Biotechnology, Instituto de Agroquímica y Tecnología de los Alimentos (CSIC), Avda. Agustín Escardino, 7, 46980 Paterna, Spain; 2Department of Science, Universidad Internacional de Valencia-VIU, Pintor Sorolla 21, 46002 Valencia, Spain

**Keywords:** wine yeast, chromosomal rearrangements, copy number variation, hybridization, horizontal gene transfer, domestication

## Abstract

Wine yeast have been exposed to harsh conditions for millennia, which have led to adaptive evolutionary strategies. Thus, wine yeasts from *Saccharomyces* genus are considered an interesting and highly valuable model to study human-drive domestication processes. The rise of whole-genome sequencing technologies together with new long reads platforms has provided new understanding about the population structure and the evolution of wine yeasts. Population genomics studies have indicated domestication fingerprints in wine yeast, including nucleotide variations, chromosomal rearrangements, horizontal gene transfer or hybridization, among others. These genetic changes contribute to genetically and phenotypically distinct strains. This review will summarize and discuss recent research on evolutionary trajectories of wine yeasts, highlighting the domestication hallmarks identified in this group of yeast.

## 1. Introduction

Since ancient times, humans have employed the ability of the yeast, mainly *Saccharomyces cerevisiae* to transform sugars into ethanol and several desirable compounds in order to produce foods and beverages of which wine and beer are the best-known products obtained from this process [1,2,3]. Somehow, the use of wine yeasts by humans in fermentative processes during millennia could be considered as an unaware directed evolution process. *S. cerevisiae* is the most extensively studied yeast belonging to *Saccharomyces* genus which is currently formed by eight species (Figure 1), some of them related to industrial environments while others are exclusively located in natural environments such as wild forest [4,5]. *Saccharomyces cerevisiae* is very prone to deal with the stresses encountered in the main industrial niches such as wine and beer, such as osmotic stress, high ethanol levels, temperature, low pH, anaerobiosis, among others [3,6,7,8,9,10,11]. *S. cerevisiae* dominance is mainly due to its huge ability to convert a plethora of sugars into ethanol and CO_2_, the latter acting as a potent antimicrobial compound to which *S. cerevisiae* is quite resistant [12]. In this sense, *S. cerevisiae* presents the ability to make and accumulate ethanol even in aerobic conditions, as a consequence of the so-called Crabtree effect [13]. This enhanced fermentative activity likely resulted from a major event in the genetic evolution of this yeast lineage, the whole genome duplication (WGD) event which enabled *S. cerevisiae* to have an increased glycolytic flux [14]. The so-called “make–accumulate–consume” strategy has been widely proposed by several authors as an ecological advantage of *S. cerevisiae* to outcompete other microorganisms [15,16] and once competitors are overcome, the ethanol can be used in aerobic respiration [17].

Nowadays, several commercial yeast strains are available for each industrial application [18]. These industrial strains are genetically and phenotypically divergent from their wild counterparts [5,19] and, with very few exceptions, industrial strains can be grouped into several lineages which correlate with their application in the industry [3,20,21,22,23,24,25,26,27,28]. Adaptation to harsh environments such as winemaking conditions can occur through smaller and/or larger genetic changes [27,29,30]. Small-scale changes including insertions, deletions and single nucleotide polymorphisms (SNPs) that may generate alteration in the gene expression and/or in the functionality of the encoded protein [12,20,31,32,33,34,35]. Larger changes such as chromosomal rearrangements (duplications, translocations, and inversions) which may alter the genomic expression through changes in the genetic environment and copy number variations (CNV) which modify the dosage of a given gene [19,36]. Furthermore, interspecific hybridization can generate new genetic combinations through introgression and horizontal gene transfer events that could not be achieved through small-scale changes such as SNPs. Indeed, SNPs are only a small proportion of the genetic changes involved in adaptation [37] likely due to the fact that adaptation to harsh niches needs to be fast in order to produce drastic changes in the phenotype, and these changes are difficult to acquire with a single nucleotide change.

In this review, we have summarized the most relevant genetic changes involved in adaptation, all of which have been reported in domesticated *Saccharomyces* genus strains.

**Figure 1 microorganisms-10-01811-f001:**
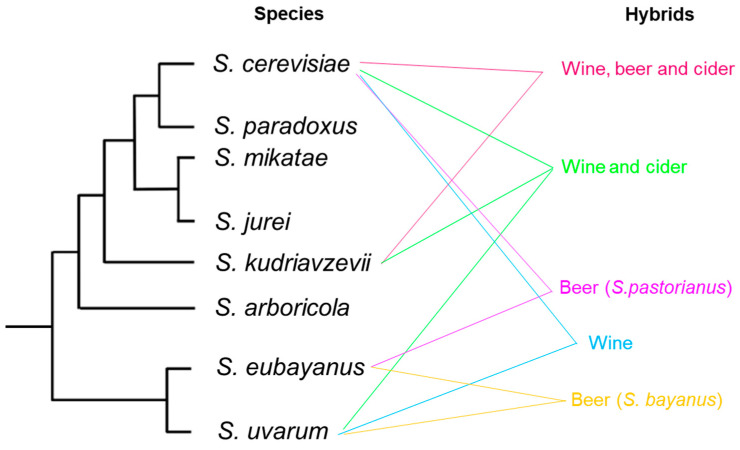
Schematic cladogram indicating phylogenetic relationships within *Saccharomyces* genus, isolation source and frequently isolated hybrids (adapted from Boynton and Greig, (2014) [38]).

## 2. Sugar Transporters (*HXT3* and *FSY1*)

The ability to consume fructose in wine yeast is crucial at the end of fermentation in order to maintain good fermentation rates and therefore to reach dryness. The transport of hexoses in *S. cerevisiae* occurs by facilitated diffusion carriers encoded by different genes, some of them belonging to HXT family [39]. In *S. cerevisiae,* from the 17 HXT genes, only seven of them (*HXT1*–*HXT7*) are needed to grow on glucose or fructose [40].

A research work in a wine yeast (Fermichamp^®^) with high fructophilic activity found an allelic variant of *HXT3* gene which is responsible for a better fructose fermentation performance [41]. *HXT3* codes for a major glucose transporter in wine fermentation, which also improves fructose consumption [42]. In this sense, other work corroborated these results and showed that all the tested strains related to sparkling wine presented the Fermichamp-like *HXT3* allele [43]. These strains are known to be very robust due to their ability to ferment both glucose and fructose even in high ethanol levels [44]. In addition, this allele is abundant in Flor yeast that are fructophilic [3,45].

Thus, the screening and selection of strains bearing this *HXT3* allele could contribute to find new high-performance strains with useful properties for industrial applications.

In the last decade, the sequencing of the wine yeast EC1118 enabled the identification of three large chromosomal regions, A, B and C acquired through horizontal gene transfer (HGT) independently from different yeast species [46]. Recently, the distant species *Torulaspora microellipsoides* has been identified as the donor source of region C [47]. This region contains 19 genes, including *FSY1*, which codes a high-affinity fructose transporter that may present an advantage when fructose concentration is higher than glucose at the end of the fermentation [48]. Recent work has shown that this region is widespread in both wine and Flor yeast [3].

In wine yeast, fructose utilization is crucial in order to maintain a high fermentative rate at the end of the process and lead the wines to dryness [41], and the strict protocols of wine yeast selection have contributed to the fixation of beneficial alleles in this environment. Flor yeast, when are in the velum form, develop an oxidative metabolism in which the main fermentable carbon source available is the fructose, therefore the acquisition of this fructophilic phenotype may be also beneficial [48,49].

## 3. Cysteine-S-β-lyase *IRC7*

*IRC7* encodes a cysteine-S-β-lyase enzyme involved in the production of thiols in wines [50,51]. This gene presents two different alleles, a complete allele (*IRC7^F^*) and a shorter version (*IRC7^S^*) that has a deletion of 38-bp and provokes the expression of a truncated gene [50]. This deletion produces a truncated protein of about 340 amino acids (aa) instead of the usual 400 aa, which generates an enzyme with lower β-lyase activity [50].

Curiously, the short version of the enzyme is widespread among commercial wine yeast [50,52,53,54]. In fact, an analysis of the prevalence of *IRC7^S^* in 223 *S. cerevisiae* strains showed that 88% of the studied strains bear the truncated *IRC7* allele [52]. Furthermore, a recent work [53] has shown that *IRC7^S^* can be also detected in other domesticated clades such as beer, while is completely absent in wild isolates [20,55]. In this regard, QTL analysis identified a new *IRC7* allele from a clinical strain as determinant for a higher production of 4-mercapto-4-methylpentan-2-one to the wine [50]. This new allele was the result of an introgression from *S. paradoxus* highlighting the huge phenotypic diversity available in nature populations of yeast that can be used in genetic improvement programs [5,38].

The reason why this truncated version of *IRC7* has been selected in the wine clade remains unclear, even more so when taking into account that the *IRC7^F^* allele is strongly related with the production of desired aromas in wine; however, this is likely related to its biological role in yeast. It has been proposed a role of *IRC7^F^* on cysteine homeostasis [56], thus a complete *IRC7* may affect the availability of the intracellular cysteine pool and consequently compromise glutathione production. Therefore, due to the paramount role of the sulphur assimilation pathway in oxidative stress protection in wine fermentation, *IRC7^s^* may have a role in this process [6,7]. This hypothesis was supported by other work in which authors observed less oxidative damage in strains harboring the short version in an oxidative-stress shock assay [53]. In addition, a recent study has identified that *IRC7* expression is repressed under high copper concentrations, thus it could have a role in copper resistance, which is a very interesting role in winemaking conditions [54].

In any case, the paradoxical distribution of the truncated version in wine yeast seems to have some evolutionary advantage in this clade because in addition to the truncated version prevalence, in those strains carrying one or two active copies of the *IRC7* gene, a high proportion (143 out of 179 wine strains) harbor inactivating SNPs [54].

## 4. Sulphite Resistance

Sulphite resistance is a paramount trait in wine yeasts due to the use of this compound in wineries as antioxidant and microbial inhibitor [8,57]. *Saccharomyces* species have evolved to adapt to the stress produced by sulphite by different mechanisms, which include an increase in acetaldehyde levels, which binds to SO_3_^2−^, the regulation of the sulphite uptake pathway, and the detoxification of sulphite through the plasma membrane pump encoded by the *SSU1* gene [8]. Wine strains are significantly more resistant to SO_2_ than other strains, mainly laboratory strains, and this is due to the evolution of a human-made environment for industrial production. Figure 2 shows the different large chromosomal changes that have been described and linked to higher expression levels of the *SSU1* gene and, therefore to higher sulphite tolerance [58,59,60]. 

To date, three different chromosomal rearrangements (CR) (i.e., VIIItXVI, XVtXVI and invXVI) have been described in *S. cerevisiae* to generate a more efficient sulphite pumping efflux and, therefore, a more sulphite resistant strains. Interestingly, these three independent evolutionary events have arisen by parallel pathways driven by the strong selective pressure exerted by alcoholic beverage producers.

The reciprocal translocation of *SSU1* gene between chromosomes VIII and XVI (native position) was identified in the wine yeast T73 in the early 2000s [58]. This rearrangement (VIIItXVI) has generated a *SSU1* allele (*SSU*-R) with higher expression levels than native *SSU1*, which produces an increase in its sulphite tolerance. The molecular mechanism governing the up-regulation of the *SSU1*-R allele is a promoter exchange with the constitutive gene *ECM34* (located in chromosome VIII) which puts the *SSU1*-R gene under the control of the *ECM34* promoter. The expression levels of *SSU1* in this reorganization was dependent of the number of acquired sequences of 76-bp repeats from the *ECM34* promoter. Furthermore, even though the Fzf1p binding sites are still present, the allele *SSU1*-R is not regulated by this transcription factor [58,61].

A later study observed other translocation between chromosomes XVI and XV (XVtXVI) that placed the *SSU1* coding region head to tail under the control of the promoter of the *ADH1* gene, which is constitutively expressed [59]. This chromosomic rearrangement was also correlated with higher *SSU1* expression levels and sulphite resistance. Both events were traditionally found only in wine yeasts: the XVtXVI translocation being the less frequent, suggesting a more recent event in the evolution [8,57].

Recently, a third different chromosomal rearrangement was identified in the commercial wine yeast ICV-GRE: an inversion of 38.5 Kb within the chromosome XVI, probably due to sequence microhomology between the regulatory regions of the genes *SSU1* and *GCR1* [60]. This inversion (invXVI) increased the sulphite tolerance of the strain to a similar level of that observed in the VIIItXVI and XVtXVI translocations [58,59]. Authors observed that the resistance to sulphite was also correlated with a higher expression of the *SSU1* gene and this expression was independent from the *FZF1* transcription factor, despite the conservation of the binding sites. The higher expression levels of *SSU1* gene in this strain were due to a combination of a part of the *GCR1* sequence upstream of *SSU1* together with the removal of part of the native promoter sequence [60].

A recent research study evaluated the prevalence of each of the three CR by using labelled primers with different fluorochromes in 586 yeast strains from different environments [62]. The translocation VIIItXVI proved to be the most abundant in the studied population while XVtXVI was the more efficient allele in terms of sulphite resistance in natural must [62].

The expression of any of the three described CR was regulated by sulphite presence in the external milieu or into the cell [61]. There is only one example of an industrial yeast strain harboring a sulphite-inducible *SSU1* allele with a putative new regulatory system [63]. In this strain, *SSU1* expression was greatly up-regulated in the presence of sulphite.

These three CR events that occurred in an independent manner integrate a hallmark on how human selection drives parallel evolutionary routes in yeast.

More recently in 2021, a new work on sulphite resistance identified for the first time a phenotypic convergence caused by independent CR in *S. cerevisiae* and *S. uvarum*, which are two of the most divergent *Saccharomyces* species [64]. Authors have identified two new translocations VII^XVI^ and XI^XVI^ that also generate an up-regulation of the *SSU1* gene, conferring higher sulphite tolerance [64]. These new CR were found in different strains of *S. uvarum*, while VIII^XVI^ was found in several European and South American strains, translocation XI^XVI^ was only observed in a European strain isolated from cider fermentation.

These results point out to the strong selective pressure cause by sulphite in human-made environments and *SSU1* gene promoter region is a hotspot for evolutionary processes at different taxonomic levels. In fact, the continuous discovering of different CR linked to sulphite tolerance in *S. cerevisiae* and more recently in *S. uvarum* could indicate that probably in other species present in the initial stage of wine fermentation such as *Hanseniospora uvarum*, *Metschnikowia pulcherrima*, *Brettanomyces sp*. among others, may exist also some sulphite detoxification mechanism. Further research on CR involving these species could be valuable and interesting.

## 5. Copper Tolerance 

The copper-based fungicides namely “Bordeaux mixture” have been used traditionally in vineyards [65]. However, copper is a toxic compound for several organisms including yeast. Yeasts have developed adaptive mechanisms in order to eliminate copper excesses, but when the intracellular concentrations are elevated can lead to cell death. To overcome this issue, metallothioneins, a class of low molecular weight molecules rich in cysteine residues are able to bind to copper ions to avoid their lethal effects [66]. In *S. cerevisiae*, the *CUP1* gene is the most representative member of this group of metallothioneins [10,67]. Wine strains usually present CNV of this gene with an increase dosage because this higher number of *CUP1* copies exert an effective protective mechanism against the copper-based fungicides. Some studies have proposed a positive correlation between higher CNV of *CUP1* gene and copper resistance as an adaptation strategy [10,22,68,69]. In a study performed in 100 strains, authors found that the copy number of *CUP1*, ranging from one to 18 copies, was associated with the phenotype of copper tolerance and CNV was responsible for ~50% of copper resistance variation in the population [22]. Other work evidenced that yeast cells exposed to a high concentration of copper amplify the copy number of the *CUP1* gene, which leads to the rapid emergence of adapted clones, and this phenomenon occurs in response to environmental copper levels due to *CUP1* transcriptional activation [70,71]. A recent study [69] analyzed the correlation between *CUP1* CNV and copper resistance in a collection of 273 *S. cerevisiae* strains from different geographical and ecological origins. A very wide range of CNV values was found ranging from zero to 79 copies. Statistical analyses highlighted the positive correlation between CNV and copper tolerance, but this association was dependent on a threshold value and this threshold was different between the geographical origins. For instance, among the vineyard strains, the Brazilian strains had a higher threshold value (copy number = 13) than the European populations (copy number = 9).

Furthermore, it has been identified a promoter variant in *CUP1* gene in the wine yeast strain EC1118 which is beneficial when dealing stress and indicates that, together with the increase in CNV, modulation of the expression of *CUP1* is another adaptation mechanism in yeast [72]. In an experimental laboratory evolution work under the presence of copper, 27 of the 34 copper-adapted lines obtained during the evolution process showed an increase in CNV of *CUP1* gene through tandem duplication or aneuploidy of chromosome VIII [73].

The presence of this variation in the number of copies of *CUP1* gene among *S. cerevisiae* strains but not in the rest of *Saccharomyces* species may point out a true event of human-driven selection or directed evolution for industrial production.

## 6. Oligopeptide Uptake: FOT Genes

Yeasts require nitrogen in order to construct some other essential molecules such as proteins and DNA and, thus, nitrogen is one of the key regulators in yeast. During fermentation, ammonium and free amino acids are the main nitrogen sources although some other secondary sources, namely oligopeptides and polypeptides, also contribute to the nitrogen levels [74]. Oligopeptide transport in yeast is carried out by several proton-coupled symporters depending on the peptide length. The proton-dependent Oligopeptide Transporter PTR2 is the best-described transporter of di- and tripeptides together with DAL5, which also transports dipeptides [75,76]. OPT1 and OPT2, Oligopeptide Transport family members, are involved in the uptake of tetra- and pentapetides (OPT1) [77,78]. The horizontally acquired Region C from *T. microellipsoides*, first identified in the wine yeast EC1118 [46], also contains the genes *FOT1*–*2* encoding for oligopeptide transporters, which considerable broaden the range of oligopeptides transported by PTR2 and DAL5 [79]. Region C, and its rearrangements, are frequent among wine yeast and, even though gene losses and conversion within FOT genes are common, they are very well conserved among the genes of the region C, indicating a possible evolutionary advantage [47,80].

Marsit et al. 2015 [47,81] showed that the presence of FOT genes provided a competitive advantage in the wine environment due to their ability to transport a wide range of oligopeptides, especially those containing glutamate, which are the most prevalent. This change in the oligopeptide diversity uptake resulted in higher viability and fermentative activity during wine fermentation. Deletion studies of both FOT genes in a haploid strain derived from EC1118 showed that the wild type (WT) strain produced higher biomass and therefore consumed more nitrogen than the mutant strain [81]. As a consequence of this higher uptake of glutamate-rich peptides, an induction of genes involved in de novo synthesis of glutathione and amino acids is produced, which led to a protection against oxidative stress [81]. Additionally, the WT strain generated a wine with better organoleptic properties due to the production of lower levels of acetate and higher amounts of ester acetates and fusel alcohols [81]. This fact was later associated with the peptides source of the must [82]. Furthermore, different FOT allele variants have been identified, generated by gene conversion from *T. microellipsoides* that present different substrate preferences. These differences in the preference and recognition of the peptides were dependent on the presence of specific amino acids in certain positions within the oligopeptide chain [83].

Besides FOT genes, several genes from the newly acquired regions present putative nitrogen-related functions [46]. Furthermore, other introgressions from *S. kudriavzeviii* and, mainly, *S. eubayanus* were observed in *S. uvarum* wine strains [84]. All the introgressed regions located in chromosome II contained *ASP1* gene encoding the cytosolic L-asparaginase that degrades asparagine to be used as a nitrogen source [85]. These genes may be an adaptive advantage taking into account that nitrogen can be limited in grape must. These results may indicate the concerted evolution of wine yeast genome associated with nitrogen utilization as recently suggested [86]. As proposed, in non-limiting conditions, efficiency enhancing mutations cannot be fixed in well-mixed populations because the individuals without these beneficial changes have equal availability of resources compared with those carrying the efficiency enhancing mutations [87]. However, in winemaking conditions where nitrogen can be limited, yeast carrying efficiency enhancing mutations have an advantage and thus, these changes can be selected in the population [86].

Little is still known about the physiological role of the FOT family in wine yeasts and more research is needed in order to unravel the oligopeptide consumption and their relationship with other metabolic pathways, which could shed light into new functionalities of these peptides in wine-related environments.

## 7. Biofilm Formation: FLO Genes

Flor strains, which are involved in sherry wine production, are able to form a biofilm on the wine surface when fermentation is finished and change their metabolism from fermentative to oxidative in the presence of ethanol and low amounts of fermentable sugar [24,88,89]. Although flor yeasts are closely related to the wine strains [90], their unique life style have rendered their genetic structure more complex. Recently, genomic analysis stated that flor yeasts are an independent clade that emerged from the wine group through a relatively recent bottleneck event [45]. The expression of FLO genes in laboratory strains is responsible for several cell wall-dependent phenotypes such as flocculation, invasion of substrate and biofilm formation [91,92].

The ability to form a biofilm is largely dependent on the acquisition of two changes in the *FLO11* gene [93], which encodes for hydrophobic cell wall glycoprotein that regulates cell adhesion, pseudohyphae, chronological aging and biofilm formation [94,95,96]. The first change was a 111-nt deletion within the promoter region of *FLO11*, which led to an increased expression [93,97]. This deletion is characteristic of Spanish, French, Italian, and Hungarian sherry strains [90].

Furthermore, *FLO11*, like the majority of the genes encoding cell wall proteins, contain intragenic tandem repeats [98]. In this regard, rearrangement in the central tandem repeat section of the ORF was responsible for producing a more hydrophobic FLO11, increasing the capacity of the yeast cells to adhere to each other [93,99]. However, the expanded *FLO11* allele present in wild flor yeasts is highly unstable under non-selective conditions [99]. Authors analyzed the ability to generate a buoyant biofilm in yeast harboring several short alleles. Curiously, biofilm formation was not correlated with the number of repeats due to certain short alleles, but not others, conferred floatability [99]. This result may indicate that not only the presence of expansions in the central domain of *FLO11* but also modifications in the ratio and/or distribution of those repetitions may be crucial in order to acquire the floatability phenotype in flor yeast.

A recent work used 142 flor stains from various countries analyzed the presence of both the 111-nt deletion and the gene expansion [90]. Authors detected the presence of the 111-nt deletion in 36 flor strains isolated from the different countries tested. Regarding the length of core region, results showed that 22 strains presented a core region, which is longer than that of wine strains.

Due to their peculiarities, flor yeasts have emerged as a potent and promising tool in order to gain insights of yeast speciation and domestication, different life-styles and some biotechnological applications such as the use of biofilms [100].

## 8. Inactivation of Aquaporin Genes

Water homeostasis is needed for osmoregulation and many other aspects of yeast life and is often associated with the presence of functional aquaporins (AQY) [101]. The loss of function of the aquaporins *AQY1* and *AQY2* represents a case of adaptive mechanism by inactivation in wine strains, resulting in an advantage fitness in high-osmolarity environments [102,103]. Several deletions or mutations, which lead to premature stop codons, have been observed in AQY genes of wine strains. Aquaporins are crucial for the survival in freeze-thaw environments, such as those founded in natural niches, however, inactivation of both genes provides an increased fitness on high-sugar substrates such as wine fermentation, which could avoid or make more difficult the migration between environmental niches promoting ecological speciation [102,104]. A recent research work on domestication trajectories in beer and wine strains has shown that the adenine 881 deletion that inactivates *AQY1* [102] was observed in the 13 wine strains tested [31]. In another study, authors observed that in a set of six cachaça strains clustered within the wine clade, all of them presented at least one inactive aquaporin gene and a high proportion of them (62%) had the two genes inactive [25]. On the contrary, Malaysian strains isolated from nectar of Bertram palm, which are not domesticated, present also non-functional alleles of aquaporin genes, highlighting that the loss of function is not strictly led by a domestication event and may have some advantages unrelated to human activity [31,32]. In fact, aquaporins have been proposed to be involved in adaptation to the surroundings, thus the selection pressure to maintain functional proteins in wild yeast, which are continuously facing different environmental stresses, may be stronger than in domesticated strains [105].

## 9. Interspecific Hybridization

Hybridization between different species is a phenomenon that occurred in all kingdoms of life, which frequently leads to infertile individuals [106]. This process has played a paramount role in the evolution of eukaryotes by generating new phenotypic diversity involved in the adaptation to new environments among divergent lineages [107,108,109].

Hybrids may have some advantages over parental strains in wine fermentation such as higher resistance to various stresses occurring during fermentation [106,110,111,112,113,114]. In this sense, *S. kudriavzevii* and *S. uvarum* are more cryotolerant compared with *S. cerevisiae* wine strains, while *S. cerevisiae* is more tolerant to high levels of ethanol [115,116,117,118,119]. Natural hybrids can exhibit interesting traits from both parentals, being able to grow under ethanol and low temperature stress [113,120] or produce other desirable aromas such as the fruity thiol 4-mercapto-4-methylpentan-2-one [121,122]. The analysis of natural hybrid of *S. cerevisiae* × *S. uvarum* isolated from wine [80] shows better performance in cold fermentations, higher glycerol production, lower acetic acid production and increased production of several interesting aroma compounds.

In the wine niche, hybrids between *Saccharomyces* genus species have been naturally found as an adaptation mechanism to human-made fermentative environments. Hybridization could be considered as an adaptation strategy due to their abundance. However, it is also plausible that the harsh environments trigger hybridization events [123]. Natural hybrids between *S. cerevisiae* and *S. kudriavzevii* are commonly found in cold-climate European countries where these hybrids even outcompete *S. cerevisiae* during fermentation due to low temperature fermentations [110,120,124,125]. *S. cerevisiae* × *S. uvarum* hybrids and the triple hybrid *S. cerevisiae* × *S. uvarum* × *S. kudriavzevii* have been naturally isolated from wine and cider environments [110,126,127]. Curiously, *S. kudriavzevii* has not been isolated from fermentative environments, thus, it remains unclear where these hybrids are generated [124,128,129].

The hybridization process is generally associated with gradual genome stabilization by which CR and changes in the genomic contribution of each parental are observed [106,124,130,131,132,133]. A recent study analyzed the genomic contribution of each parental in 23 natural *S. cerevisiae* × *S. kudriavzevii* hybrids when growing at low temperature [134]. Authors showed that hybrids with the better fitness in cold where those with a higher expression of the cryophilic *S. kudriavzevii* alleles of cold stress markers. Furthermore, it has also been observed that mitochondrial inheritance is also crucial because hybrids with an *S. kudriavzevii* mitochondrial genome are able to maintain a larger proportion of the *S. kudriavzevii* genome portion than hybrids with an *S. cerevisiae* mtDNA [135]. In fact, another study has shown that the genomes of *S. cerevisiae × S. kudriavzevii* hybrids were mainly characterized by the maintenance of the *S. cerevisiae* subgenome and a progressive reduction of the *S. kudriavzevii* proportion [131]. VIN7, a commercial strain which is an almost complete allotriploid hybrid *S. cerevisiae* (2*n*) × *S. kudriavzevii* (*n*), shows several recombination and genomic changes between the genomes of both parental species [136].

Besides the huge biotechnological impact of hybrids, recent advances in yeast genomics and the development of new methodologies have opened up our understanding about the hybridization potential as a source of genetic diversity [137,138,139,140,141,142]. The diversity offered by the hybrid population has the potential to be used in quantitative genetics analysis in order to unveil the structure of complex traits [143].

## 10. Conclusions

Under detrimental conditions such as those found during wine fermentation, yeast populations need to promote adaptive responses in order to survive in this harsh environment. Wine yeast has developed a plethora of genetic mechanisms to adapt to the wine niche, including variation of the copy number of certain genes, structural rearrangements, HGT or interspecific hybridization. From this set of genomic adaptations, structural variations leading to duplications of genes, chromosomic segments, full chromosomes or even the complete genome are the major cause of adaptation [133,144]. Besides its great impact by changing dosage, this structural variation can be the substrate for evolution and even the generation of new genes [145]. Indeed, several research works of several microorganisms experimentally evolved may confirm that this kind of structural variation is frequent and can lead, eventually, to adaptation [37,55,73,146,147,148,149]. This phenomenon may be a fast solution in order to adapt quickly to harsh environments before other SNPs and CR could alleviate the potential detrimental effect or trade-offs caused by the initial genomic change [150,151,152]. In wine yeast, the genomic events leading to an increase of CNV such as in copper tolerance as well as the gain of new genetic material through hybridization or HGT are the most abundant from those identified to date.

Great efforts have been made in the last few years to expand the whole-genome sequence dataset of yeast strains isolated from both wild and industrial environments [27,153]. This fact together with the successful application of several approaches such as QTL mapping, computational models, optical mapping or phenomics, among others, in order to identify variants involved in niche adaptation, have increased our knowledge of the genotype–phenotype associations [34,154,155,156,157,158,159,160,161,162,163,164,165,166]. Furthermore, increase the number and diversity of sequenced strains as well as the use of new long-reads sequencing platforms will provide a better picture of the population structure and evolutionary history of *Saccharomyces* and the extent of domestication in this process. Furthermore, the development of these new techniques will help us to identify CR and HGT related to human-driven selection.

A better understanding and identification of the variants underlying the domestication process will also increase the possibility of use this new knowledge to select better-adapted yeast to specific industrial environments.

## Figures and Tables

**Figure 2 microorganisms-10-01811-f002:**
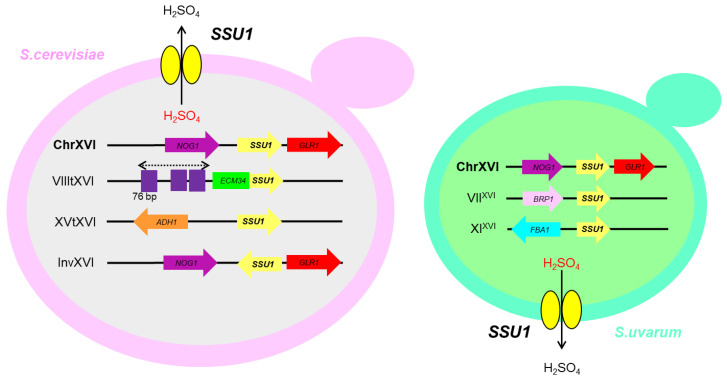
Schematic view of chromosomal rearrangements identified in sulphite-resistant strains of *S. cerevisiae* and *S. uvarum* species. VIIItXVI was generated by the homology among the promoters of *ECM34* and *SSU1* genes. Several 76-bp (in purple) repetitions were found in the promoters, together with a positive correlation between the number of 76-bp repeats and sulfite resistance. XVtXVI involves the ADR1 and FZF1 binding sites of the promoter of *ADH1* and *SSU1* genes, respectively. InvXVI was produced by microhomology between the sequences of the regulatory regions of the genes *SSU1* and *GCR1* genes. The translocations found in S. uvarum, VIIXVI and XIXVI, are likely mediated by microhomology regions between *BRP1*-*SSU1* and *FBA1*-*SSU1*, respectively.

## Data Availability

Not applicable.

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
