# Peer review of "Genomic Adaptations of Saccharomyces Genus to Wine Niche"

_microorganisms, 2022, doi:10.3390/microorganisms10091811_

Round 1

Reviewer 1 Report

The authors have summarized in this bibliographic revision (microorganisms-1881045) named “Genomic adaptation to anthropic niches within Saccharomyces genus: the wine environment” the Saccharomyces genomics data related to yeast cellular adaptation to wine niches. This specific bibliographic review was written in a manner that conforms to standard scientific documents. In the manuscript, a reasonable number of references are cited as evidence of substantive background. The results in general are presented and explain in a clear manner with due reference to scientific published articles, however the discussions sections proposed by the authors are not as challenge as is expected in this kind of documents. Although the review is quite well written and presented, it does contain a few errors which may need to be addressed by the authors before the manuscript is deemed acceptable for publication in "Microorganisms".

Minor points

This reviewer does not understand why the scientific concept “directed evolution” it is not used in this document, the subject referred by the authors is a good example of that.

This reviewer suggests eliminating the redundancy of the proposed title by a shorter one like “Genomic adaptation of Saccharomyces genus to anthropic wine niches”.

Page 2 The first paragraph must be changed. The SNP concept is a quite recent concept in yeast genetics research. Many authors, before the wide used of this concept, already published genetic adaptations of Saccharomyces strains to industrial niches, which showed big genomic rearrangements like aneuploidy, genome duplication, genetic deletions, etc. The SNPs might help to explain the differences between Saccharomyces cerevisiae strains adapted to a non-standard environmental condition as wine yeast.

Page 3 The authors clearly explain the putative function of IRC7s on nitrogen metabolism vs. oxidative stress tolerance. But the authors said that remain unclear the selection of IRC7 truncated version in wine industry, which is logic because it could reduce thiols concentration in fermented beverages, but what about a putative yeast toxicity effect of thiols in presence of ethanol, have the authors some information from this point of view?

Page 3 This reviewer suggest change “S. cerevisiae cells” by “Saccharomyces species”

Figure 2 only have the title, the authors must enclose a brief explanation to help readers to understand it, example the means of VIIItXVI, etc.

Page 4 The name S.cerevisiae must be changed by S. cerevisiae.

Page 4: For this reviewer, it is hard to consider at the “human selection” when it refers to microorganisms during non-recombinant DNA biotechnology periods. Perhaps, it is clearer “events have arisen by parallel pathways to the selective pression mediate by alcoholic beverages makers”.

Page 4 The authors must enclose the name of the protein encoded by the cited genes as ADH1, GCR1, etc. Comment valid for the entire document.

Page 6 The authors must use the established standard for S. cerevisiae to name proteins that must be in normal capital letter (PTR2, DAL5, OPT1 and OPT2) and to refer to genes must be in italic capital letter, PTR2. Commentary valid for the entire manuscript.

Page 6 The expression “that nitrogen is limiting in grape must” must be changed by “that nitrogen can be limiting in grape must”.

Page 7, the authors mention in the other phenotype sections the validations of the phenotypes by technologies of recombinant DNA. This reviewer does not understand why they do not keep this pattern for FLO11.

Page 7, “functional aquaporins (AQY)” must be change by “functional aquaporins (AQY)”.

Page 7 The authors should be more specific in expressions like “the complete set of wine strains tested”, how many strains means a set? It must be change by “a set of n wine strains”. This commentary is valid for the entire document.

Page 7 Perhaps the authors can comment the origin of the isolated Malaysian strains without functional AQY.

Page 7 The sentence “Humans have been …”, must be eliminated does not bring any information to the reader, on the other hand, Triticale was created in a laboratory and your review focus in spontaneous genetic changes in wine yeast.

Page 8 The sentence “Wine yeast employ …”  must be changed by “Wine yeast has developed a plethora ...”.

The references must be written in only one standard; example: ref. 5 “Exploring the Use of Wild Strains …” why the authors use capital letters in this title? ref. 6 “Saccharomyces cerevisiae” it must be in italic, etc.

Author Response

Reviewer 1

The authors have summarized in this bibliographic revision (microorganisms-1881045) named “Genomic adaptation to anthropic niches within Saccharomyces genus: the wine environment” the Saccharomyces genomics data related to yeast cellular adaptation to wine niches. This specific bibliographic review was written in a manner that conforms to standard scientific documents. In the manuscript, a reasonable number of references are cited as evidence of substantive background. The results in general are presented and explain in a clear manner with due reference to scientific published articles, however the discussions sections proposed by the authors are not as challenge as is expected in this kind of documents. Although the review is quite well written and presented, it does contain a few errors which may need to be addressed by the authors before the manuscript is deemed acceptable for publication in "Microorganisms".

Minor points

This reviewer does not understand why the scientific concept “directed evolution” it is not used in this document, the subject referred by the authors is a good example of that.

We consider that directed evolution is more related to the lab environment in which the researcher drives the evolution to gain some specific traits in the yeast strain. However, we are explaining in the document different adaptive mechanisms as consequence of the use of wine yeasts by man in fermentative processes during millennia. This process could be considered an unaware directed evolution.

Attending to the reviewer suggestion, we have now introduced the next sentence in introduction:  Somehow, the use the use of wine yeasts by man in fermentative processes during millennia could be considered as an unaware directed evolution process

This reviewer suggests eliminating the redundancy of the proposed title by a shorter one like “Genomic adaptation of Saccharomyces genus to anthropic wine niches”.

We appreciate reviewer’s suggestion and the title has been changed accordingly as follows: “Genomic adaptations of Saccharomyces genus to wine niche”

Page 2 The first paragraph must be changed. The SNP concept is a quite recent concept in yeast genetics research. Many authors, before the wide used of this concept, already published genetic adaptations of Saccharomyces strains to industrial niches, which showed big genomic rearrangements like aneuploidy, genome duplication, genetic deletions, etc. The SNPs might help to explain the differences between Saccharomyces cerevisiae strains adapted to a non-standard environmental condition as wine yeast.

This section has been rewritten in order to clarify the concepts:

“Adaptation to harsh environments such as winemaking conditions can occurred through smaller and/or larger genetic changes [28,30,31]. Small-scale changes including insertions, deletions and single nucleotide polymorphisms (SNPs) that may generate alteration in the gene expression and/or in the functionality of the encoded protein [12,20,32–37]. Larger changes such as chromosomal rearrangements (duplications, translocations, and inversions) which may alter the genomic expression through changes in the genetic environment and copy number variations (CNV) which modify the dosage of a given gene [19,38]. Furthermore, interspecific hybridization can generate new genetic combinations through introgression and horizontal gene transfer events that could not be achieved through small-scale changes such as SNPs”

Page 3 The authors clearly explain the putative function of IRC7s on nitrogen metabolism vs. oxidative stress tolerance. But the authors said that remain unclear the selection of IRC7 truncated version in wine industry, which is logic because it could reduce thiols concentration in fermented beverages, but what about a putative yeast toxicity effect of thiols in presence of ethanol, have the authors some information from this point of view?

Unfortunately we have not been able to find any publication correlating thiols toxicity in presence of ethanol and the distribution of the truncated IRC7 in our extensive literature search. However it is an interesting hypothesis to test because it easily explained why the truncated version has been selected in the wine clade.

Page 3 This reviewer suggest change “S. cerevisiae cells” by “Saccharomyces species”

We have corrected this statement.

Figure 2 only have the title, the authors must enclose a brief explanation to help readers to understand it, example the means of VIIItXVI, etc.

We thank reviewer’s suggestion and the legend of figure 2 has been explained better.

“Figure 2. Schematic view of chromosomal rearrangements identified in sulphites resistant strains of S. cerevisiae and S. uvarum species. VIIItXVI was generated by the homology among the promoters of ECM34 and SSU1 genes. Several 76-bp (in purple) repetitions were found in the promoters, together with a positive correlation between the number of 76-bp repeats and sulfite re-sistance. XVtXVI involves the ADR1 and FZF1 binding sites of the promoter of ADH1 and SSU1 genes, respectively. InvXVI was produced by microhomology between the sequences of the regula-tory regions of the genes SSU1 and GCR1 genes. The translocations found in S. uvarum, VIIXVI and XIXVI, are likely mediated by microhomology regions between BRP1-SSU1 and FBA1-SSU1, respectively.”

Page 4 The name S.cerevisiae must be changed by S. cerevisiae.

We have amended the mistake.

Page 4: For this reviewer, it is hard to consider at the “human selection” when it refers to microorganisms during non-recombinant DNA biotechnology periods. Perhaps, it is clearer “events have arisen by parallel pathways to the selective pression mediate by alcoholic beverages makers”.

The reviewer is right and we have modified the statement accordingly.

Page 4 The authors must enclose the name of the protein encoded by the cited genes as ADH1GCR1, etc. Comment valid for the entire document.

We have revised the document and changed all the names by standard names of genes and proteins.

Page 6 The authors must use the established standard for S. cerevisiae to name proteins that must be in normal capital letter (PTR2, DAL5, OPT1 and OPT2) and to refer to genes must be in italic capital letter, PTR2. Commentary valid for the entire manuscript.

We have revised the document and changed all the names by standard names of genes and proteins.

Page 6 The expression “that nitrogen is limiting in grape must” must be changed by “that nitrogen can be limiting in grape must”.

We have amended the mistake.

Page 7, the authors mention in the other phenotype sections the validations of the phenotypes by technologies of recombinant DNA. This reviewer does not understand why they do not keep this pattern for FLO11.

We appreciate reviewer’s comment and we have improved this section.

“Furthermore, FLO11, like the majority of the genes encoding cell wall proteins, contain intragenic tandem repeats [101]. With this regard, rearrangement in the central tandem repeat section of the ORF was responsible of produce a more hydrophobic FLO11, increasing the capacity of the yeast cells to adhere to each other [96,102]. However, expanded FLO11 allele present in wild flor yeasts is highly unstable under non-selective conditions [102]. Authors analyzed the ability to generate a buoyant biofilm in yeast harboring several short alleles. Curiously, biofilm formation was not correlated with the number of repeats due to certain short alleles, but not others, conferred floatability [102]. This result may indicate that not only the presence of expansions in the central domain of FLO11 but also modifications in the ratio and/or distribution of those repetitions may be crucial in order to acquire the floatability phenotype in flor yeast.

A recent work used 142 flor stains from various countries analyzed the presence of both the 111-nt deletion and the gene expansion [93]. Authors detected the presence of the 111-nt deletion in 36 flor strains isolated from the different countries tested. Regarding the length of core region, results showed that 22 strains presented a core region, which is longer than that of wine strains.

Due to their peculiarities, flor yeast have emerged as a potent and promising tool in order to gain insights of yeast speciation and domestication, different life-styles and some biotechnological applications such as the use of biofilms [103]. ”

Page 7, “functional aquaporins (AQY)” must be change by “functional aquaporins (AQY)”.

We have amended the mistake.

Page 7 The authors should be more specific in expressions like “the complete set of wine strains tested”, how many strains means a set? It must be change by “a set of n wine strains”. This commentary is valid for the entire document.

We thank reviewer’s appreciation and we have revised the data to include the specific numbers of each study.

Page 7 Perhaps the authors can comment the origin of the isolated Malaysian strains without functional AQY.

We have now introduced the isolation source of these strains.

Page 7 The sentence “Humans have been …”, must be eliminated does not bring any information to the reader, on the other hand, Triticale was created in a laboratory and your review focus in spontaneous genetic changes in wine yeast.

The reviewer’s is right and we have deleted this sentence from the manuscript.

Page 8 The sentence “Wine yeast employ …”  must be changed by “Wine yeast has developed a plethora ...”.

We have modified the sentence.

The references must be written in only one standard; example: ref. 5 “Exploring the Use of Wild Strains …” why the authors use capital letters in this title? ref. 6 “Saccharomyces cerevisiae” it must be in italic, etc.

We thank reviewer’s appreciation, it was a mistake in the reference manager and we have now corrected all the references.

Reviewer 2 Report

The review is rather interesting and surely informative. I find some aspects that, however, need further consideration: 

1. The title, some but not all the mechanisms described derive from anthropogenic pressure. Most of them are adaptive to stressing conditions typical of wine and must. I guess that the title should be changed accordingly. 

2. The review is an enumeration of papers dealing with various topics. In this sense it is easy to follow and nice to read. However, it lacks the "meat" of the authors and risks to become a bibliographic report, rather than a critical review. 

3. According to the n.2 point, there are sections in which the authors make a statement and give the reference, without a discussion or a link to other aspects or simply an explanations. One such case is of  the end of paragraph 5 (citation 80)

4. The concluding remarks could have been an opportunity to gather all the info in some consistent way, for instance by type of genetic phenomenon occurring, but it is not so. This makes the review informative, but really not so exciting to read

Author Response

Reviewer 2

The review is rather interesting and surely informative. I find some aspects that, however, need further consideration: 

  1. The title, some but not all the mechanisms described derive from anthropogenic pressure. Most of them are adaptive to stressing conditions typical of wine and must. I guess that the title should be changed accordingly. 

We appreciate reviewer’s suggestion and the title has been changed accordingly as follows: “Genomic adaptations of Saccharomyces genus to wine niche”

  1. The review is an enumeration of papers dealing with various topics. In this sense it is easy to follow and nice to read. However, it lacks the "meat" of the authors and risks to become a bibliographic report, rather than a critical review. 

Following reviewer’s suggestion, we try to improve the discussion of the complete manuscript and we have introduced new paragraphs in each section (highlighted in yellow).

  1. According to the n.2 point, there are sections in which the authors make a statement and give the reference, without a discussion or a link to other aspects or simply an explanations. One such case is of  the end of paragraph 5 (citation 80)

The reviewer’s is right and the section has been completed as follows:

“As proposed, in non-limiting conditions efficiency enhancing mutations cannot be fixed in well-mixed populations because the individuals without these beneficial changes have equal availability of resources comparing with those carrying the efficiency enhancing mutations [90]. However, in winemaking conditions where nitrogen can be limited, yeast carry efficiency enhancing mutations have an advantage and thus, these changes can be selected in the population [89].

Little is still known about the physiological role of the FOT family in wine yeasts and more research is needed in order to unravel the oligopeptide consumption and their relationship with other metabolic pathways, which could shed light into new functionalities of these peptides in wine-related environments.”

  1. The concluding remarks could have been an opportunity to gather all the info in some consistent way, for instance by type of genetic phenomenon occurring, but it is not so. This makes the review informative, but really not so exciting to read

We thank the constructive comment and we have introduced more information and references in the concluding remarks section in order to improve the discussion.

“Under detrimental conditions such as those found during wine fermentation, yeast populations need to promote adaptive responses in order to survive in this harsh environment. Wine yeast has developed a plethora of genetic mechanisms to adapt to wine niche, including variation of the copy number of certain genes, structural rearrangements, HGT or interspecific hybridization. From this set of genomic adaptations, structural variations leading to duplications of genes, chromosomic segments, full chromosomes or even the complete genome are the major cause of adaptation [136,147]. Besides its great impact by changing dosage, this structural variation can be the substrate for evolution and even the generation of new genes [148]. Indeed, several research work of several microorganisms experimentally evolved may confirm that this kind of structural variation is frequent and can lead, eventually, to adaptation [39,57,75,149–152]. This phenomenon may be a fast solution in order to adapt quickly to harsh environments before other SNPs and CR could alleviate the potential detrimental effect or trade-offs caused by the initial genomic change [153–155]. In wine yeast, the genomic events leading to an increase of CNV such as in copper tolerance as well as the gain of new genetic material through hybridization or HGT are the most abundant from those identified to date. 

Great efforts have been made in the last years to expand the whole-genome sequence dataset of yeast strains isolated from both wild and industrial environments [28,156]. This fact together with the successful application of several approaches such as QTL mapping, computational models, optical mapping or phenomics, among others, in order to identify variants involved in niche adaptation have increased our knowledge of the genotype–phenotype associations [36,157–169]. Furthermore, increase the number and diversity of sequenced strains as well as the use of new long-reads sequencing platforms will provide a better picture of the population structure and evolutionary history of Saccharomyces and what is the extent of domestication in this process. Furthermore, the development of these new techniques will help us to identify CR and HGT related to human-drive selection.

The better understanding and identification of the variants underlying domestication process will also increase the possibility of use this new knowledge to select better-adapted yeast to specific industrial environments.”

Reviewer 3 Report

Authors have written a very nice review, compiling information regarding adaptation of Saccharomyces yeasts to wine environments. Review is very well written and the bibliographic search is remarkable, even though, in my opinion, some very important papers were left out.

I recommend paper acceptance, after some important citations are included in the text, as explained below. Without them, authors cannot claim that they reviewed the relevant bibliography in the field.

- page 2, line 1-5 - authors need to cite also the following works developed by the Portuguese team:

10.1186/s12864-017-3816-1

10.1371/journal.pone.0066523

10.1093/femsyr/fov063

- Aquaporins section - authors missed an important citation from a Sweden group:

doi.org/10.1042/BC20040144

- Concluding remarks:

* when concluding about other techniques, such as QTL, authors need to expand this paragraph, and include other modern applications such as the ones published by:

      - 10.1016/j.febslet.2009.03.068

      -10.1002/yea.3016

     - 10.1091/mbc.E15-07-0466

Minor comments:

- citations 3 and 10 are duplicated 

- many italics are missing in the references list

Author Response

Reviewer 3

Authors have written a very nice review, compiling information regarding adaptation of Saccharomyces yeasts to wine environments. Review is very well written and the bibliographic search is remarkable, even though, in my opinion, some very important papers were left out.

I recommend paper acceptance, after some important citations are included in the text, as explained below. Without them, authors cannot claim that they reviewed the relevant bibliography in the field.

- page 2, line 1-5 - authors need to cite also the following works developed by the Portuguese team:

10.1186/s12864-017-3816-1

10.1371/journal.pone.0066523

10.1093/femsyr/fov063

We thank reviewer’s suggestion and we have included now in this section the proposed works.

- Aquaporins section - authors missed an important citation from a Sweden group:

doi.org/10.1042/BC20040144

We thank reviewer’s suggestion and we have included now in this section the proposed work.

- Concluding remarks:

* when concluding about other techniques, such as QTL, authors need to expand this paragraph, and include other modern applications such as the ones published by:

      - 10.1016/j.febslet.2009.03.068

      -10.1002/yea.3016

     - 10.1091/mbc.E15-07-0466

Following reviewer’s recommendation, we have improved this paragraph with new information and references including those proposed by the reviewer.

“Great efforts have been made in the last years to expand the whole-genome sequence dataset of yeast strains isolated from both wild and industrial environments [28,156]. This fact together with the successful application of several approaches such as QTL mapping, computational models, optical mapping or phenomics, among others, in order to identify variants involved in niche adaptation have increased our knowledge of the genotype–phenotype associations [36,157–169].”

Minor comments:

- citations 3 and 10 are duplicated

We have amended the mistake.

- many italics are missing in the references list

We have modified and corrected the reference list.

Round 2

Reviewer 2 Report

The paper has been amended in many ways and I think it could be published. However, I noticed among the  tracked changes some inconsistency of the language, for instance a " can occurred" at pag. 2 or sometimes the gene names without italic.  I must admit that I do not feel well when reading a track change text and I refrain from making a list of such problems since I am not so sure. Still, I recommend to be careful .